# Antioxidant and Antimicrobial Properties of Hydrolysed Collagen Nanofibers Loaded with Ginger Essential Oil

**DOI:** 10.3390/ma16041438

**Published:** 2023-02-08

**Authors:** Mariana Daniela Berechet, Carmen Gaidau, Aleksandra Nešić, Rodica Roxana Constantinescu, Demetra Simion, Olga Niculescu, Maria Daniela Stelescu, Irina Sandulache, Maria Râpă

**Affiliations:** 1The National Research & Development Institute for Textiles and Leather, 16 Lucretiu Patrascanu Street, 030508 Bucharest, Romania; 2Faculty of Technology, University of Novi Sad, 21102 Novi Sad, Serbia; 3Faculty of Materials Science and Engineering, Polytechnic University of Bucharest, 060042 Bucharest, Romania

**Keywords:** collagen hydrolysate, ginger essential oil, nanofiber, electrospinning, 3D structure, antibacterial and antifungal activities

## Abstract

Hydrolysed collagen obtained from bovine leather by-products were loaded with ginger essential oil and processed by the electrospinning technique for obtaining bioactive nanofibers. Particle size measurements of hydrolysed collagen, GC-MS analysis of ginger essential oil (EO), and structural and SEM examinations of collagen nanofibers loaded with ginger essential oil collected on waxed paper, cotton, and leather supports were performed. Antioxidant and antibacterial activities against *Staphylococcus aureus* and *Escherichia coli* and antifungal activity against *Candida albicans* were also determined. Data show that the hydrolysed collagen nanofibers loaded with ginger EO can be used in the medical, pharmaceutical, cosmetic, or niche fields.

## 1. Introduction

The approaching of adopting naturally derived polymers in the form of a combination with plant extracts such as essential oils is preferred when it is desired to find the biological properties of the constituent compounds in medicinal products or cosmetics or they are used in the substitution or regrowth of human tissue [1]. The excellent biocompatibility and biodegradability of natural polymers versus that of synthetic polymers is an important topic to consider in the domain of medical, cosmetic, or biomedical engineering products [2].

The mixing together of natural polymers, for example, collagen, with essential oils to take advantage of both materials can lead to the most effective antifungal, antimicrobial, and regeneration results [3,4]. Collagen is a major element in the extracellular matrix (ECM) and, therefore, the largest plentiful protein in human beings [5]. Its important biological features when compared with different polymers lead to a best selection for medical and cosmetic applications [6]. In various research studies, the collagen composite fibres have been used in combination with polycaprolactone (PCL) and the addition of highly toxic solvents, such as hexafluoroisopropanol [4,7,8]. The technological process during the manufacturing and extraction of collagen might have an important effect on solubility, viscosity, and subsequent processability by electrospinning [9,10,11]. Electrospinning is a nano-technique that uses an electrostatic spinning machine as a tool to make nanofibers simply and easily. Thanks to the convenience of nanofibers, such as light fabrication, easy operation, high surface-area-to-volume ratio, and surface bionics of the extracellular matrix for suitability in biomedical engineering, the electrospun nanofiber scaffolds have been extensively investigated in tissue engineering [12,13,14,15,16,17,18], drug delivery [5,19,20], energy science [21,22,23], sensors [24,25,26], catalysts [27,28], filtration [29,30], food science [31,32,33], and environmental science [34]. Electrospinning technology has been tested for the manufacturing of collagen nanofibers extracted in acidic medium from calfskin [35], or collagen from bovine dermis [36], collagen extracted in alkaline medium from fish skin [37], and the extraction in acidic medium from collagen in fish skin in the presence of PCL [38]. Due to the high efficiency of capturing bioactive substances and ideal protection for bioactivity [39], electrospinning nanofibers can be examined for controlled release of bioactive and food compounds [40,41]. Electrospinning could play important roles in maximizing the use of collagen resources extracted from fish skin, cowhide, or rabbit skin for tissue engineering and drug and cosmetic administration [18,21,37]. The use of the electrospinning technique allows the production of nanofiber materials with high ratios between surface and volume as well as the loading of bioactive compounds with applications in medical products, cosmetics, and medicines [12,14,15,16,17,18]. Various polymer types have been used by electrospinning for the fabrication of active materials containing encapsulated active agents. This emerging technology was also employed to deposit matt fibres on cotton to improve its performance [13,16]. They could even be designed to obtain products with regenerative or antioxidant, antimicrobial, or antifungal protective properties used in medicines and cosmetics [42].

The objective of this research is to obtain nanofibers from hydrolysed collagen loaded with ginger essential oil. Through the electrospinning technique, nanofibers with 3D structure and antibacterial and antifungal properties were obtained and collected on waxed paper, cotton cloth, and leather supports. The use of ginger essential oil loaded in hydrolysed collagen can be a natural alternative to the pathogen resistance of antimicrobial agents [43].

## 2. Materials and Methods

### 2.1. Chemical Materials

In this study, the following chemical materials were used: ginger (*Zingiber officinale*) essential oil from Adams Vision SRL (Targu Mures, Romania), sodium carbonate (Na_2_CO_3_) from SG GXG Chemicals SRL (Budesti-Racovita, Valcea, Romania), Alcalase 2.4 L (protease from *Bacillus licheniformis*, 2.4 U/g) from Sigma-Aldrich (St. Louis, MI, USA), hydrated calcium oxide (CaO) purchased from Cristal R Chim SRL (Bucharest, Romania), and pearl sodium hydroxide (NaOH) acquired from Lachner (Neratovice, Czech Republic).

### 2.2. Obtaining of Hydrolysed Collagen Concentrates

To obtain the hydrolysed collagen, the bovine leather was alkaline hydrolysed in 400% (*w*/*v*) water in the presence of 10% CaO (*w*/*w*) for 4 h at a temperature of 80 °C and under traditional mechanical stirring. Hydrolysis was continued enzymatically with 0.4% Alcalase (*w*/*w*) with pH adjustment at 8–9, with a solution of 20% concentration NaOH (*v*/*v*) for 3 h under mechanical stirring at a temperature of 65 °C. The enzyme deactivation occurred at a temperature of 80 °C for 15 min. The collagen hydrolysate thus obtained was decanted and filtered and further concentrated at a ratio of 1:4 in a rotary evaporator at a temperature of 65 °C, 150 rpm, and pressure of 150 mbar. Finally, a concentrated hydrolysed collagen of honey consistency and amber colour was obtained [44].

### 2.3. Physical–Chemical Characterization of Hydrolysed Collagen

The concentrated hydrolysed collagen was analysed by physical–chemical methods according to the standards in force and methods such as ash content (EN ISO 4047:2008), dry matter (EN ISO 4684:2006), total nitrogen and protein (ISO 5397:1996), pH (STAS 8619/3:1990), amino nitrogen (according to ICPI method), and electrical conductivity (EN ISO 27883:1997). The viscosity of concentrated hydrolysed collagen hydrolysate was determined with a Brookfield AMETEK DV2T TC-550 Viscometer (Brookfield, Toronto, ON, Canada) at a temperature of 25 °C. The particle size and zeta potential of collagen were evaluated by using a Dynamic Light Scattering (DLS) instrument from Malvern (Zetasizer Nano-ZS, Malvern Hills, UK).

### 2.4. Characterization of Ginger Essential Oil (EO)

#### 2.4.1. GC-MS Analysis

Bioactive compounds from ginger EO were identified using Gas Chromatography coupled with Mass Spectroscopy (DSQ II MS, Thermo Scientific, Waltham, MA, USA), supplied with TR-5 MS non-polar capillary column having dimensions of 60 m × 0.25 μm × 0.25 μm. The technical parameters temperature, heating rate, temperature of split/splitless injector, and pneumatic control system of pressure/carrying gas flow were fixed in the range between room temperature to 350 °C, 0.1 °C/min to 120 °C/min, 50 °C to 375 °C, and 5%, respectively.

#### 2.4.2. Determination of Total Phenolic Content (TPC) and DPPH Free-Radical Scavenging Assay

Determination of total phenolic content (TPC) of ginger EO was carried out based on Folin-Ciocalteu’s method [44]. A concentration of 60 mg mL^−1^ ginger EO was achieved in ethanol at room temperature, and 50 µL of the ginger EO sample prepared as above was mixed with 0.240 mL Folin-Ciocalteu’s reagent and 3.6 mL distilled water. The obtained solution was maintained for 5 min in a dark place at ambient temperature, and then 0.68 mL 7.5% (*w*/*v*) solution of sodium carbonate (Na_2_CO_3_·10H_2_O) was added. This solution was moderately stirred and kept for 30 min in darkness at a temperature of 40 °C. The solution absorbance was read at a wavelength of 740 nm against blank, and the TPC calculated as mg of gallic acid equivalent per g of dry weight (GAE/g dw) was carried out using a calibration curve based on gallic acid.

Antioxidant activity of ginger EO was performed by mixing together 2.5 mL of DPPH (1,1-diphenyl-2-picrylhydrazyl) in an ethanol solution (0.150 mmol/L) with 0.5 mL of 60 mg mL^−1^ ginger EO. After 30 min standing in a dark place at ambient temperature, the absorbance was read at 517 nm by using a UV/VIS spectrometer (Orion UV-VIS AQUAMATE 8000, Thermo Fisher Scientific, Eindhoven, The Netherlands). Every analysis was conducted in triplicate.

### 2.5. Obtaining of Nanofibers Based on Hydrolysed Collagen and Ginger EO

The concentrated hydrolysed collagen together with 10% ginger EO was mechanically stirred at 400 rpm for 50 min. Nanofibers based on the above homogenized solution were fabricated by electrospinning process (Fluidnatek equipment, Bioinicia, Valencia, Spain). Nanofibers of hydrolysed collagen were also obtained, as a control. The applied electric potential, the flow rate, and the distance between the injector needle and the storage support fixed on the metal collector were adjusted depending on the support used. The electrospinning processing parameters were as follows: an electrical potential of 9.8 kV applied for waxed paper, 18.5 kV for cotton, and 20.6 kV for leather between the positive injector needle and negative support; a flow rate of 550 µL for waxed paper, 700 µL for cotton, and 1000 µL for leather; and a distance of 13 cm between injector needle to each support type. The experiments were carried out at room temperature and a relative humidity of 32 ± 5%.

### 2.6. Ginger EO Release

Ginger EO loading efficiency was determined by ultraviolet-visible (UV-VIS) spectroscopy. About 20 mg of electrospun material was solubilized into ethanol (96%). A volume of supernatant liquid was achieved by centrifugation at 6000 rpm for 10 min, then filtration (0.45 μm polytetrafluoroethylene (PTFE) filter, Whatman, Maidstone, UK) and, in the end, a wavelength of 290 nm was examined. The theoretical loading capacity of ginger EO was determined taking into account the essential oil introduced in the electrospinning solution, in other words, 10 wt.% directly proportional to the volume of hydrolysed collagen solution. Calibration curves containing ginger EO concentrations ranging from 0 mg mL^−1^ to 20 mg mL^−1^ were used by UV-VIS testing of known concentrations of ginger EO in ethanol.

### 2.7. Attenuated Total Reflectance (ATR)—Fourier Transform Infrared (FT-IR) Spectroscopy

ATR-FT-IR technique was performed with an FT-IR/ATR spectrometer—Jasco 4200 (Jasco Corporation, Tokyo, Japan)—operating in the wavenumber ranging from 4000 cm^−1^ to 650 cm^−1^, which detected the absorbance peaks at 0.5 cm^−1^ spectral resolution. The device was equipped with a Michelson interferometer with incident angle of 45°, with Digital Signal Processing technology (DSP), cubic angular mirrors with self-aligning mechanisms, and a standard thermostatic DLATGS Peltier detector (Jasco Corporation, Tokyo, Japan).

### 2.8. Scanning Electron Microscopy (SEM)

The electronic microscopic examinations were created using an FEI Quanta 200 microscope (FEI, Valley City, ND, USA), which operated at high vacuum, 15 kV, in addition to a gaseous secondary electron (GSED) detector.

### 2.9. Microbiological Analyses

An adapted method for determining antibacterial and antifungal activity was used according to the standard ISO 20743:2013—Determination of antibacterial activity of textile products. This document mentions quantitative test methods for antibacterial activity assessment of all antibacterial textile items, which include nonwovens. Both nanofibers based on collagen (control) and containing ginger EO were analysed towards *Escherichia coli* (Gram negative bacteria) ATCC 11229, *Staphylococcus aureus* ATCC 6538 (Gram positive bacteria), and *Candida albicans* ATCC 10231 yeast. The cell concentration in the inoculum used was of 2.4 × 10^4^ CFU/mL (Colony Forming Units) for *Escherichia coli*, 3.6 × 10^4^ CFU/mL for *Staphylococcus aureus*, and 2.8 × 10^4^ CFU/mL for *Candida albicans*. The initial cell concentration was calculated earlier by decimal dilutions (10^−4^) using sterile deionized water; then, from the last dilution, a volume of 100 µL was taken for each strain and spread further on Nutrient Agar. The plate counting was recorded at 24 h of incubation, this being kept as a reference for the cellular developments of the control. Antibacterial activity ratio (R) was calculated according to Equation (1).
(1)$$R\left(\%\right)=\frac{C_{t}-T_{t}}{C_{t}}\times 100$$
where: *C_t_* is the average number of colonies of two control samples after 24 h or the specified incubation period, expressed as CFU/mL; *T_t_* is the average number of colonies of two test samples after 24 h or the specified incubation period, expressed as CFU/mL. The results were also indicated as the logarithmic reduction resistance against bacteria and fungi. The antimicrobial efficacy and degree of microbial and logarithmic reduction for each sample were estimated depending on the initial cell concentration.

## 3. Results

### 3.1. Physical–Chemical Parameters of Hydrolysed Collagen

The concentrated hydrolysed collagen was characterized by determining dry matter, ash, total nitrogen, protein content, pH, aminic nitrogen, and electrical conductivity properties. Table 1 shows the obtained values.

The hydrolysed collagen extracted enzymatically shows high contents of protein (82.43%) and dry matter (60.40%) and a viscosity value (1623 cP) suitable for electrospinning.

### 3.2. DLS Analysis

The size of collagen particles measured possess an average of 1150 nm with majority populations at 109.3 nm (52.3%), 9.7 nm (22.3%), and 1.9 nm (14.3%) (Figure 1) and a zeta potential value of −7.64 mV (Figure 2), as presented in Table 2.

The particle size measurement of hydrolysed collagen showed small particles, with 36.6% of particle sizes below 10 nm and 52.3% of particle sizes at 109.3 nm, meaning an easy penetration of tissues. The high polydispersity of hydrolysed collagen with high particle size and low particle size components is due to the associative properties of proteins in high concentration (60.4%).

Other authors reported a polydisperse solution of hydrolysed collagen, with diverse particle size from 65 nm to 246 nm [45], or from a low value up to around 5000 nm [46]. The largest particle (~5500 nm), which is monodispersed and different from the other three smaller groups with broader distribution, could be due to the insufficient concentration of alkali used during the synthesis process of collagen [46].

The zeta potential value is low corresponding to a non-stable dispersion.

### 3.3. Gas Chromatography—Mass Spectroscopy (GC-MS)

Table 3 reveals the retention times, percentage of area in which the compounds were found, as well as and names.

The ginger EO obtained by distilling ginger rhizomes is used in aromatherapy due to its invigorating action, in perfumery bringing warm, woody notes, and also in cosmetics, due to its anti-inflammatory, peripheral circulation stimulation, and toning properties [47,48,49,50]. The strong antimicrobial, antifungal, and antioxidant activities of ginger EO are well recognized [51]. Ginger essential oil contains monoterpenic hydrocarbons (α-pinene, camphene, and myrcene), oxygenated compounds (citral, cineol, and citronellol acetate), and sesquiterpenoids (zingiberene) [47,48,52]. The composition of essential oils depends on environmental factors for plant growth: light intensity and duration, altitude, harvest season, soil, and nutrients [49,50].

The majority compounds identified in ginger EO were limonene (21.88%), camphene (21.47%), α-pinene (11.29%), cineole (10.46%), and zingiberene (9.32%) (Table 3).

### 3.4. Efficiency of Ginger EO in Collagen Nanofibers

It was demonstrated that the amount of ginger EO from hydrolysed collagen nanofibers was 32 mg mL^−1^, below the theoretical concentration of 100 mg mL^−1^. Accordingly, the loading efficiency was 32%. Berechet et al. [44] also reported a 29–39% efficiency of encapsulation during the nanoencapsulation of thyme and oregano EO into collagen. It was expected that the efficiency of ginger EO loading was higher immediately after the electrospinning process and it decreased due to the volatility of EO.

### 3.5. Total Phenolic Content (TPC) and DPPH Free-Radical Scavenging Activity Assessment

Nanofibers loaded with ginger EO indicated a high capacity to prevent the DPPH free radicals of 79.8 ± 7% and 0.88 mg GAE/g dry matter. This outcome is generally in accordance with that obtained by Ali et al. [53]. Thus, the antioxidant activity by DPPH for the extract of ginger rhizome in petroleum ether and chloroform: methanol was 97.47 ± 0.93% and the total phenolic content was 60.34 ± 0.43 mg GAE/g dry matter. In another paper, Wijayanti et al. [54] found 4.85 mg GAE/g dry weight in the case of ginger extraction in methanol. However, it was revealed that the phenolic compounds commonly occur as part of the action to defend against invading pathogens or in plant response to adverse environments, and they are not precisely associated with the growth activity and progress of plant tissue [53].

### 3.6. Structural Characterization by ATR-FT-IR

The ATR-FT-IR spectra for the ginger EO and nanofibers based on hydrolysed collagen loaded with ginger EO and hydrolysed collagen (control) are shown in Figure 3. The maximum absorption of ginger EO appeared at the characteristic bands (Figure 3a), as follows: 3478 cm^−1^ (–OH groups from phenols), 2918 cm^−1^ (asymmetric stretching vibration of –CH_2_ group), 1680 cm^−1^ (stretching vibration of –C=O), 1453 cm^−1^ (–C–H group), 1376–1082 cm^−1^ (–C–O–C group), 982 cm^−1^ (C–C group), and 982–883 cm^−1^ (C–C–H) [50,52].

The characteristic peaks detected for nanofibers based on hydrolysed collagen containing ginger EO appear at 3266 cm^−1^ (O–H and N–H stretching vibration), 1632 cm^−1^ (amide I and C=O stretching), 1544 cm^−1^ (amide II and N–H bending), 1444–1411 cm^−1^ (CH_2_ bending), 1239 cm^−1^ amide III (CN and NH), and 1084 cm^−1^ (stretching of the C–O group). The spectrum for nanofibers based on hydrolysed collagen containing ginger EO is not different than that for hydrolysed collagen nanofibers. The incorporation of ginger EO did not lead to changes in absorption peaks, but to a high intensity of absorption peaks as compared with control (Figure 3b). This behaviour could be explained by the low amount of ginger EO introduced as well as to the possible volatilization of bioactive compounds in time. A similar effect was reported by Al-Hilifi [55], who incorporated ginger EO into chitosan. Figure 3c shows the main absorption peaks for ginger EO and hydrolysed collagen nanofibers in the fingerprint region of 1800–800 cm^−1^. The overlapping of characteristic peaks can be observed in the spectrum of nanofibers containing ginger EO, meaning that the interactions between hydrolysed collagen and ginger EO components took place.

### 3.7. SEM Examination

Figure 4, Figure 5 and Figure 6 reveal the SEM images observed for nanofibers of hydrolysed collagen and hydrolysed collagen with ginger EO collected on waxed paper, cotton, and leather supports at a magnification of 8000×.

The collagen nanofibers loaded with ginger essential oil obtained by electrospinning have a 3D structure with average sizes of 524.3 nm to 665.5 nm, increased by ginger essential oil loading (Figure 4, Figure 5 and Figure 6) as compared to collagen nanofibers with average diameters of 464.2 nm to 531.2 nm on waxed paper, cotton, and leather supports (Table 4).

### 3.8. Microbiological Evaluation

Table 5 and Table 6 show the antibacterial and antifungal activities performed for hydrolysed collagen nanofibers and hydrolysed collagen nanofibers containing ginger EO towards *Escherichia coli*, *Staphylococcus aureus*, and *Candida albicans*.

Antibacterial tests show better results of collagen nanofibers containing ginger EO with resistance against *Escherichia coli* and *Staphylococcus aureus* of 73.33% and 96.8%, respectively, as compared with control collagen nanofibers, which show 70% resistance against *Escherichia coli* and 96.67% resistance against *Staphylococcus aureus*. It is common knowledge that essential oils are more efficient against Gram positive bacteria than Gram negative bacteria due to their different membrane structures. Each component of ginger EO has a specific ability to disrupt or penetrate the bacteria structure, directly influencing the antimicrobial activity [56]. Among all compounds of ginger EO identified by GC-MS analysis, Ar-Curcumene has an aromatic nucleus in its structure. The literature describes the antimicrobial activity of ginger EO, which is mainly attributed to the camphene, phellandrene, zingiberene, and zingerone compounds [55,57]. In addition, collagen from nanofiber compositions brings potential antibacterial activity [58,59].

The antifungal tests showed a 95.51% resistance against *Candida albicans* for hydrolysed collagen nanofibers containing ginger EO. The control (hydrolysed collagen nanofibers) recorded a 76.67% resistance against *Candida albicans.* The microbiological tests showed that the hydrolysed collagen nanofibers loaded with ginger EO have a resistance against Gram negative bacteria *Escherichia coli* (73.33%) and a higher resistance against Gram positive bacteria *Staphylococcus aureus* (96.8%) and *Candida albicans* fungus (95.51%).

Through these experiments, hydrolysed collagen nanofibers loaded with ginger EO were fabricated through the electrospinning technique and deposited on waxed paper, cotton, and leather supports. Microbiological analyses showed antibacterial and antifungal properties for nanofibers based on hydrolysed collagen containing ginger EO and could recommend its use in the manufacture of linings for gloves, hats, or shoes with a therapeutic purpose, as well as therapeutic stockings or other niche products. It would also be possible to obtain footwear that, when used for a long time and in demanding conditions for the foot (sports shoes or boots), could prevent the development of mycoses [60]. However, other tests that demonstrate the rate of change of collagen fibres under the influence of elevated temperature and moisture as well as removing ginger EO from the environment under the influence of moisture must be undertaken. In this study, the use of natural compounds for producing eco-friendly nanofibers with antifungal and antibacterial effects could be a solution for other products obtained by synthesis [61,62].

## 4. Conclusions

Hydrolysed collagen was extracted by alkaline-enzymatic hydrolysis and concentrated in a rotary evaporator to a viscosity value of 1623 cP, when it was able to be processed by electrospinning. Nanofibers based on hydrolysed collagen containing 10% ginger EO deposited on leather, textile, and waxed paper supports by the electrospinning process showed slight antimicrobial properties as compared to collagen-based nanofibers (control) against *Escherichia coli*, and *Staphylococcus aureus*. A high efficiency to the *Candida albicans* fungus was observed for the tested nanofibers loaded with ginger EO. The nanofibers fabricated by electrospinning have a well-defined 3D structure with average diameters ranging from 464.4 nm to 665.5 nm. Collagen nanofibers loaded with ginger EO could be used in medicine, pharmaceuticals, cosmetics, or niche products, as well as for natural treatment of cotton or leather products.

For future studies, the investigations of water absorption and the release of bioactive compounds to the environment will be conducted, so that the results of this work can be transferred to other natural polymers loaded with essential oils that are processed by the electrospinning technique.

## Figures and Tables

**Figure 1 materials-16-01438-f001:**
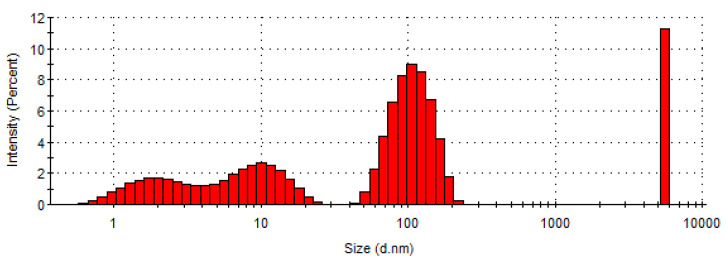
Histogram of particle size distribution in concentrated hydrolysed collagen.

**Figure 2 materials-16-01438-f002:**
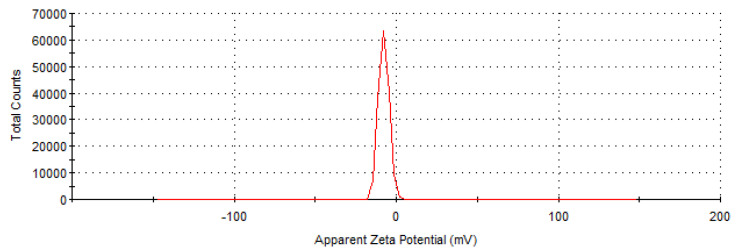
Zeta potential measurement of concentrated hydrolysed collagen.

**Figure 3 materials-16-01438-f003:**
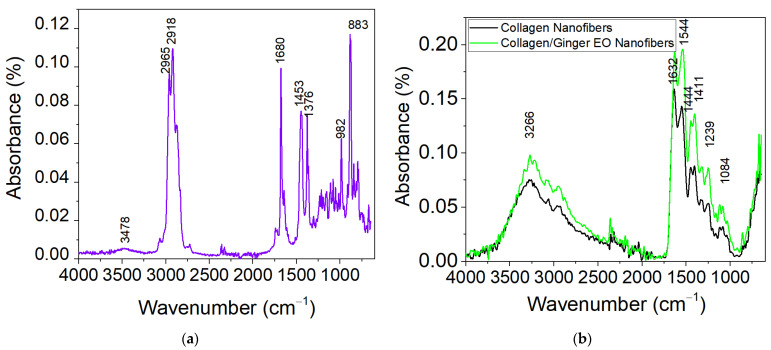
ATR FT-IR spectra for ginger EO (**a**), nanofibers of hydrolysed collagen and hydrolysed collagen loaded with ginger EO from 4000–600 cm^−1^ (**b**), and nanofibers of hydrolysed collagen and hydrolysed collagen loaded with ginger EO from 1800–800 cm^−1^ (**c**).

**Figure 4 materials-16-01438-f004:**
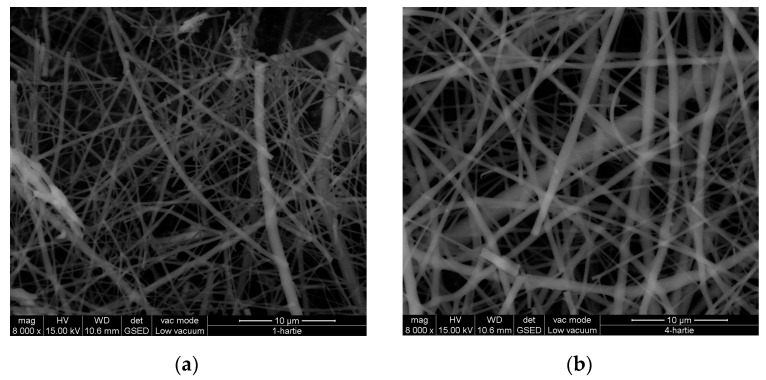
SEM images for hydrolysed collagen nanofibers (**a**) and hydrolysed collagen. Nanofibers containing ginger EO (**b**) collected on waxed paper support, at 8000×.

**Figure 5 materials-16-01438-f005:**
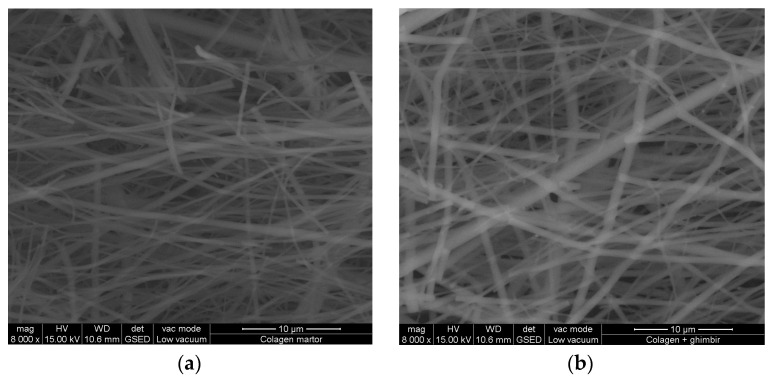
SEM images for hydrolysed collagen nanofibers (**a**) and hydrolysed collagen. Nanofibers containing ginger EO (**b**) collected on cotton support, at 8000×.

**Figure 6 materials-16-01438-f006:**
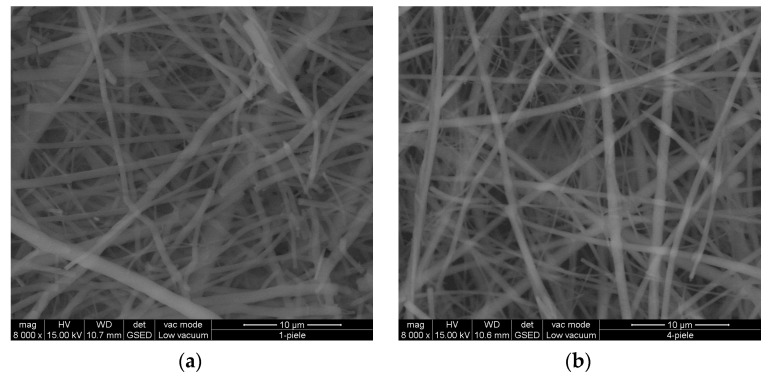
SEM images for hydrolysed collagen nanofibers (**a**) and hydrolysed collagen. Nanofibers containing ginger EO (**b**) collected on leather support, at 8000×.

**Table 1 materials-16-01438-t001:** Physical and chemical parameters of the hydrolysed collagen.

Parameters, U.M.							
Dry Matter, %	Ash ^a^, %	Total Nitrogen ^a^, %	Protein ^b^, %	pH, Units of pH	Aminic Nitrogen ^b^, %	Viscosity, cP	Electrical Conductivity, µS/cm
60.40	6.24	14.67	82.43	8.54	1.43	1623	870

^a^ Values reported on a dry matter basis, ^b^ value reported on protein basis.

**Table 2 materials-16-01438-t002:** Particle sizes and zeta potential measurements.

Sample	Main Populations						Average, nm	Pdl	Zeta Potential, mV
	Size, nm	%	Size, nm	%	Size, nm	%			
Hydrolysed collagen	1.9	14.3	9.7	22.3	109.3	52.3	1150	0.898	−7.64

**Table 3 materials-16-01438-t003:** Chemical composition of compounds identified in ginger EO recognized by GC-MS.

No. Peak	Retention Time, min.	Name of Compounds	Formula	Percentage of Area, %
1	12.99	Tricyclene	C_10_H_16_	4.34
2	13.44	α-Pinene	C_10_H_16_	11.29
3	14.27	Camphene	C_10_H_16_	21.47
4	15.78	Myrcene	C_10_H_16_	2.24
5	17.92	Limonene	C_10_H_16_	21.88
6	18.14	Cineole (Eucalyptol)	C_10_H_18_O	10.46
7	25.88	Isoborneol	C_10_H_18_O	1.40
8	30.45	Neral	C_10_H_16_O	3.89
9	32.26	Citral	C_10_H_16_O	4.03
10	36.31	Citronellol acetate	C_12_H_22_O_2_	1.35
11	42.95	Ar-Curcumene	C_15_H_22_	2.32
12	43.53	Zingiberene	C_15_H_24_	9.32
13	44.1	γ-Muurolene	C_15_H_24_	3.32
14	44.91	α-Funebrene	C_15_H_24_	2.69

**Table 4 materials-16-01438-t004:** Average diameter for hydrolysed nanofibers based on collagen and collagen containing ginger EO deposited on different supports.

Sample	Average Diameter, nm		
	Paper Waxed Support	Cotton Support	Leather Support
Nanofibers of hydrolysed collagen	464.2	485.2	531.2
Nanofibers of hydrolysed collagen containing ginger EO	665.5	524.3	649.7

**Table 5 materials-16-01438-t005:** Antibacterial activity of hydrolysed collagen nanofibers and hydrolysed collagen nanofibers loaded with ginger EO against *Escherichia coli* and *Staphylococcus aureus*.

0	Value, CFU/mL	R, %	Log_10_ Red.
*Escherichia coli*			
Inoculum concentration	T_0_ = 2.4 × 10^4^		
Nanofibers of hydrolysed collagen	T_24_ = 7.2 × 10^2^	70	0.52
Nanofibers of hydrolysed collagen loaded with ginger EO	T_24_ = 6.4 × 10^2^	73.33	0.57
*Staphylococcus aureus*			
Inoculum concentration	T_0_ = 3.6 × 10^4^		
Nanofibers of hydrolysed collagen	T_24_ = 1.2 × 10^2^	96.67	1.48
Nanofibers of hydrolysed collagen loaded with ginger EO	T_24_ = 11.6 × 10^2^	96.8	1.49

**Table 6 materials-16-01438-t006:** Antifungal activity for nanofibers based on hydrolysed collagen and hydrolysed collagen containing ginger EO against *Candida albicans*.

Sample	Value, CFU/mL	R, %	Log_10_ Red.
Inoculum concentration	T_0_ = 2.8 × 10^4^	76.67%	0.63
Hydrolysed collagen nanofibers	T_24_ = 5.8 × 10^3^	95.51%	1.35
Hydrolysed collagen nanofibers loaded with ginger essential oil	T_24_ = 1.25 × 10^3^		

## Data Availability

Data supporting reported results can be found from the corresponding authors.

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
