# Peer review of "Antioxidant and Antimicrobial Properties of Hydrolysed Collagen Nanofibers Loaded with Ginger Essential Oil"

_materials, 2023, doi:10.3390/ma16041438_

Round 1
Reviewer 1 Report
1. The text in rows 196-198 is repeated further in rows 203-206.
2. Table 4 shows the Average size of collagen nanofibers, nm, what does it mean? The diameter of the fibers, their length or average cross-sectional size?
3. It is noted in the Сonclusions. .... The nanofibers obtained by electrospinning have a well-defined 3D structure with average size of 464.4 nm to 665.5 nm.
Such a statement is not very correct. On the basis of SEM images, a non-woven material was obtained on the surface of paper or textile. This non-woven material really has a three-dimensional structure. Only one characteristic is given for the fibers, and it is necessary to specify whether it is the diameter of the fibers, average cross-sectional size or their length.
4. The keywords of the abstract do not fully reflect the content of the article, in particular, electrospinning is not mentioned as the main method of obtaining the nanofibers.
5. p.3.6. Structural Analysis by ATR-FTIR
The obtained results are given, but are not discussed in any way and no conclusions are drawn.
6. The authors of the article note that …. The objective of this research is to obtain advanced natural treatments with antibacterial and antifungal bioactivity and to confer an added value to cotton fabrics and leather by adding bioactive properties to improve the quality of life and, thus, to access new markets with the obtained products (rows 68-70 of the text).
…But it is not clear from the Сonclusions whether the goal of the research was achieved.
7. In the Conclusions it is noted "…Nanofibers based on collagen hydrolysate and collagen hydrolysate loaded with 10% ginger essential oil obtained by electrospinning on waxed paper support showed slight antimicrobial properties as compared to collagen based nanofibers against Escherichia coli, Staphylococcus aureus and Candida albicans" (rows 329-332 of the text).
It is not indicated whether the nanofibers obtained by electrospinning on textile or leather support showed antimicrobial properties.
At the end of the article, the authors note (rows 311-317 of the text).
"…Microbiological analyzes have shown antibacterial and antifungal properties and the possibility of being used in the manufacture of linings for gloves, hats or shoes with a therapeutic purpose, as well as therapeutic stockings or other niche products. It would also be possible to obtain footwear that, used for a long time and in demanding conditions for the foot (sports shoes, boots), could prevent the development of mycoses".
In my opinion, this recommendation is not very correct.
It is known that the internal parts of shoes are affected by moisture, sweat, and high temperature. Under the influence of these factors for a long time, the essential oils of ginger introduced into the composition of collagen nanofibers will be washed out and evaporated. The article does not contain data on the rate of change of collagen fibers under the influence of elevated temperature and moisture. The article does not contain data on the possibility of removing ginger essential oils into the environment under the influence of moisture. Accordingly, it is not clear how long these antimicrobial properties will be preserved.
Moreover, the authors themselves in paragraph 3.4. indicate that after the electrospinning process there is a decrease in the amount of volatile substances of ginger essential oils.
Author Response
We thank you for the valuable comments and suggestions.
- Thank you for your observation. The text in rows duplicate was removed.
-
The average size of collagen nanofibers is referred to the diameter measurement. Table 4 was corrected, accordingly.
-
We corrected the Conclusions with “average diameter”.
-
We added “electrospinning” to keywords.
-
The text was modified, as follows.
“The characteristic peaks detected for nanofibers based on hydrolysed collagen containing ginger EO appear at 3266 cm-1 (O–H and N–H stretching vibration), 1632 cm-1 (amide I and C=O stretching), 1544 cm-1 (amide II and N–H bending), 1444-1411 cm-1 (CH2 bending), 1239 cm-1 amide III (CN and NH) and 1084 cm-1 (stretching of the C–O group). Spectrum for nanofibers based on hydrolysed collagen containing ginger EO is not different than that for collagen hydrolysate nanofibers. Incorporation of ginger EO did not lead to changes in absorption peaks, but to a high intensity of absorption peaks as compared with control (Figure 3b). This behavior could be explained by the low amount of ginger EO introduced as well as to the possible volatilization of bioactive compounds in time. Similar effect was reported by Al-Hilifi [56] when incorporated ginger EO into chitosan. Figure 3c shows the main absorption peaks for ginger EO and hydrolysed collagen nanofibers in fingerprint region of 1800 - 800 cm-1. It could be observed in the spectrum of nanofibers containing ginger EO the overlapping of characteristic peaks, meaning that the interactions between hydrolysed collagen and ginger EO components taken place.”
-
We modified the objective of this paper. Indeed, obtaining of natural treatments with antibacterial and antifungal bioactivity and to confer an added value to cotton fabrics and leather by adding bioactive properties to improve the quality of life and, thus, to access new markets with the obtained products could be a possible perspective.
-
The Conclusions were completed with following text: “Nanofibers based on hydrolysed collagen containing 10% ginger EO deposited on textile and waxed paper supports by electrospinning process showed slight antimicrobial properties as compared to collagen-based nanofibers (control) against Escherichia coli, Staphylococcus aureus and Candida albicans. A high efficiency to Candida albicans fungus was observed for tested nanofibers loaded with ginger EO.”
Our idea related to the possible use of developed nanofibers for the footwear is based on the recognized fungicidal activity of ginger EO, as well as the good antifungal activity of nanofibers against Candida albicans (95.51%) compared with this property for collagen nanofibers (76.67%). We totally agree that other tests are necessary to support this affirmation.
We added the following text:
“However, other tests that demonstrate the rate of change of collagen fibers under the influence of elevated temperature and moisture, as well as removing ginger EO into the environment under the influence of moisture must be undertaken”.
Reviewer 2 Report
This is an interesting manuscript. However, it requires some corrections and explanations.
1. What components of ginger essential oil (EO) are responsible for antimicrobial properties of collagen nanofibers modified with EO ? Which components of EO contained phenolic structures ? Is gallic acid (GA) one of components of EO ? Results OF FTIR analysis do not provide sufficient evidence for presence of GA in EO. GA was not detected in EO by GC-MS method.
It seems that proteins from collagen may exhibit antimicrobial activity as well. Please check in a literature.
2. How collagen nanofibers were deposited on wax paper, cotton and leather ?
3 A symbol R described as resistance against bacteria and fungi (in subchapter 3.8) should be further explained and defined.
4. A text in lines 196-198 is repeated in lines 203-205.
5. A term "CG-SM" in line 213 should be corrected.
Author Response
We thank you for the valuable comments and suggestions.
1. We added the following text:
“Each component of ginger EO has a specific ability to disrupt or penetrate the bacteria structure, directly influencing the antimicrobial activity [57]. Among all compounds of ginger EO identified by GC-MS analysis, Ar-Curcumene has an aromatic nucleus in its structure. In the analyzed ginger EO, we did not identify gallic acid. Among all compounds of ginger EO identified by GC-MS analysis, Ar-Curcumene has an aromatic nucleus in its structure. Literature describe the antimicrobial activity of ginger EO, which is mainly attributed to the camphene, phellandrene, zingiberene, and zingerone compounds [58],[56]. In addition, collagen from nanofiber compositions brings potential antibacterial activity [59],[60].”
In the analyzed ginger EO, we did not identify gallic acid.
2. The concentrated hydrolysed collagen together with 10% ginger EO were mechanically stirred at 400 rpm for 50 min. Nanofibers based on the above homogenised solution were fabricated by electrospinning process (Fluidnatek equipment, Bioinicia, Spain). Nanofibers of hydrolysed collagen were also obtained, as a control. The applied electric potential, the flow rate and the distance between the injector needle and the storage support fixed on the metal collector were adjusted depending on the support used. The electrospinning processing parameters were: an electrical potential applied of 9.8 kV for waxed paper, 18.5 kV for cotton and 20.6 kV for leather between the positive injector needle and negative support, a flow rate of 550 µL for waxed paper, 700 µL for cotton and 1000 µL for leather, and a distance of 13 cm between injector needle to each support type. The experiments were carried out at room temperature and a relative humidity of 32 ± 5%.
3. The text was modified accordingly.
Antibacterial activity ratio (R) was calculated according to Eq. 1. (as attached)
where: Ct is the average number of colonies of two control samples after 24 h or the specified incubation period, expressed as CFU/mL; Tt is the average number of colonies of two test samples after 24 h or the specified incubation period, expressed as CFU/mL.
4. Thank you for your correction! We deleted the repeated lines.
5. Thank you for your observation! We corrected the term.

Reviewer 3 Report
Berechet et al. presented a method to incorporated ginger essential oil into collagen hydrolysate nanofibers for antioxidative and antimicrobial applications. The conclusions of this manuscript could be supported by the results. However, deeper explanations and clearer descriptions of the obtained data should be done. My comments and suggestions are as follows,
1. In 3.2. DLS analysis, the authors observed four popular particle sizes with different orders of magnitude for the collagen hydrolysate, an explanation on this huge size deviation should be provided. Also, the largest particle (~ 5500 nm) was nearly monodispersed which is quite different from the other three smaller groups with broader distribution, please provide sufficient analysis on this.
2. In 3.6. Structural Analysis by ATR-FTIR, the spectra can be better presented if the authors plot them into one graph. The fingerprint region with wavenumber below 500 cm-1 can be removed due to background noise. The spectra should be normalized with respect to certain invariant peak for better comparison.
3. In 3.7. SEM Examination section, the increase in nanofiber size on cotton support was only approx. 40 nm which was much smaller compared to paper waxed support of ~ 200 nm. Please provide explanation on this observation. Also, will the average nanofiber size influence their antimicrobial properties? The authors are encouraged to conduct more characterization on this.
4. In page 5, line 213, it should be “GC-MS” instead of “CG-SM”.
Author Response
We thank you for the valuable comments and suggestions.
1. The text was completed accordingly.
“Other authors reported a polydisperse solution of hydrolysed collagen, with diverse particle size from 65 nm to 246 nm [45], or from low value up to around 5000 nm [46]. The largest particle (~ 5500 nm), which is monodispersed and different from the other three smaller groups with broader distribution could be due to the insufficient concentration of alkali used during synthesis process of collagen [46].
The high polydispersity of hydrolysed collagen with high particle sizes and low particle sizes components is due to the associative properties of proteins in high concentration (60.4%).”
2. Thank you for the suggestions. New spectra with the fingerprint region between 4000 to 600 cm-1 were provided. The text was completed as follows.
“The characteristic peaks detected for nanofibers based on hydrolysed collagen containing ginger EO appear at 3266 cm-1 (O–H and N–H stretching vibration), 1632 cm-1 (amide I and C=O stretching), 1544 cm-1 (amide II and N–H bending), 1444-1411 cm-1 (CH2 bending), 1239 cm-1 amide III (CN and NH) and 1084 cm-1 (stretching of the C–O group). Spectrum for nanofibers based on hydrolysed collagen containing ginger EO is not different than that for hydrolysed collagen nanofibers. Incorporation of ginger EO did not lead to changes in absorption peaks, but to a high intensity of absorption peaks as compared with control (Figure 3b). This behavior could be explained by the low amount of ginger EO introduced as well as to the possible evaporation of bioactive compounds in time. Similar effect was reported by Al-Hilifi [56] when incorporated ginger EO into chitosan. Figure 3c shows the main absorption peaks for ginger EO and hydrolysed collagen nanofibers in fingerprint region of 1800 - 800 cm-1. It could be observed in the spectrum of nanofibers containing ginger EO the overlapping of characteristic peaks, meaning that the interactions between hydrolysed collagen and ginger EO components taken place.”
3. Optimal working parameters used for obtaining nanofibers were different for each support used (waxed paper, cotton, leather). Thus, for the electric potential applied between the injector needle (+) and the metal collector (-), on which each of the three types of support was fixed in turn, the following values ​​were used: 9.8 kV for waxed paper, 18.5 kV for cotton and 20.6 kV for leather; the flow rate of the collagen hydrolysate homogenized with essential ginger oil was as follows: 550 µL for waxed paper, 700 µL for cotton and 1000 µL for leather; and the distance between the injector needle and the support on which nanofibers were deposited was 13 cm for all types of support. We consider that these different optimal parameters for each support used influence the diameter of the nanofibers obtained. The type of support was different and it has different porosities that could influence the diameter of the nanofibers obtained.
We think that the antimicrobial properties are influenced by the size dimension of nanofibers. At low size dimension of nanofibers the contact with antimicrobial agents is higher as compared with similar nanofibers characterized by high size dimension. This issue should be investigated by the authors, in detail, in further papers.
4. Thank you. The change was made.